# The Use of Durum Wheat Oil in the Preparation of *Focaccia*: Effects on the Oxidative Stability and Physical and Sensorial Properties

**DOI:** 10.3390/foods11172679

**Published:** 2022-09-02

**Authors:** Francesca Vurro, Carmine Summo, Giacomo Squeo, Francesco Caponio, Antonella Pasqualone

**Affiliations:** Department of Soil, Plant and Food Science (DISSPA), University of Bari Aldo Moro, via Amendola 165/a, I-70126 Bari, Italy

**Keywords:** flat bread, durum wheat oil, acidic composition, tocotrienols, tocopherols, volatile compounds, texture profile analysis, sensorial properties

## Abstract

Durum wheat oil is an innovative oil that could be considered the “second life” of durum wheat milling by-products. In this study, we proposed the use of this oil in the reformulation of a traditional Italian greased flat bread, namely *focaccia*, whose typical sensorial features are due to the presence of relevant amounts of oil in its formulation. The chemical, physical, and sensorial features of *focaccia* with durum wheat oil (DWO) were compared with those of *focaccia* prepared with olive oil (OO) and sunflower oil (SO). The results showed the prevalence of polyunsaturated fatty acids in DWO, followed by SO. DWO was more resistant to oxidation than SO (induction time 86.2 and 66.3 min, respectively), due to its higher content of tocotrienols (1020 and 70.2 mg/kg in DWO and SO, respectively), but was less resistant than OO, richer in monounsaturated fatty acids, and contained phenolic compounds. The volatile oxidation markers, namely hexanal and nonanal, were less prevalent in OO and DWO than in SO. Texture and color were positively influenced by the use of durum wheat oil, allowing the nutritional improvement of this flat bread in a sustainable and circular manner.

## 1. Introduction

Bread is a traditional staple food consumed by people worldwide, with hundreds of different examples [1]. Among them, flat breads are the oldest, and they are still very popular due to their high versatility, organoleptic properties, and convenience [2,3]. These reasons justify the relevant growth of their market, especially in relation to the modern style of life and new eating preferences [4].

Italy has a long tradition of garnished flat breads, some of which are also renowned abroad, such as pizza. *Focaccia* is another typical garnished flat bread widely consumed in several Italian regions, where it originated [5]. *Focaccia* is oven-baked in a low pan and, prior to cooking, is topped with fresh tomato and oregano; onions and potatoes; cheese; or salt and rosemary, etc., providing a myriad of diverse and nuanced varieties [6]. This old and traditional food product, which has been included in the list of Italian Traditional Agri-Food Products (TAP) [7], is similar to pizza but has some distinct characteristics [2,5]. The consumption patterns of *focaccia* and pizza are different: the first is a quick snack consumed at any time [5], while the second is usually preferred for dinner or lunch (except for the “*pizza a portafoglio*”, the typical street food of Naples) [8]. The difference between pizza and *focaccia*, however, is not limited to their consumption patterns, as their formulation, appearance, and sensory characteristics are also different. The preparation of pizza and *focaccia* starts in the same way, by kneading flour, water, yeast, and salt. Then, only in the preparation of *focaccia*, abundant oil is incorporated into the dough and poured onto its surface to confer the typical greasiness [2,5]. Conversely, in the preparation of the “Traditional Specialty Guaranteed” (TSG) Neapolitan pizza, only a very small amount of oil is used [9]. Specifically, only extra virgin olive oil can be used in TAP-labeled *focaccia* and TSG-labeled pizza, which are high-quality niche products. However, most of the commonly marketed *focaccia* and pizza, which are not labeled as TAP or TSG, contain olive oil or sunflower oil.

Pizza has been the object of several investigations aimed at improving its nutritional features [10,11,12,13], without neglecting gluten-free versions [14,15], but very few studies are available on *focaccia*. The reformulation of *focaccia* using *Apulian black chickpea* flour, which provides anthocyanins and increases antioxidant activity, was investigated by Pasqualone et al. [12], while Delcuratolo et al. [16,17] evaluated the role of *focaccia* toppings on the oxidation stability and content of polar compounds arising from triacylglycerol oxidation and hydrolysis, which are responsible for negative health implications [18]. However, only a single study has investigated the use of fat replacers to reduce the oil content of *focaccia* [19], and no studies have compared the effects of different vegetable oils on its nutritional and sensory characteristics.

Italy is not only famous for pizza and *focaccia*, but also for pasta and special baked goods made of durum wheat semolina [20,21,22]. However, the milling process for obtaining semolina involves the production of by-products (bran, germ, and various middlings) [23], which should be upcycled and reintroduced into the food system to comply with the principles of a circular economy [24,25]. These by-products have a proven potential for oil extraction [25]. A previous study has evaluated the effect of using durum wheat oil in the preparation of biscuits [26], whose long shelf life can be affected by rancidity onset. The substitution of sunflower oil with durum wheat oil significantly increased the resistance of biscuits to oxidation due to the abundance of tocols in durum wheat oil, especially tocotrienols [26].

At this historical moment, the war in Ukraine is causing problems for the supply of sunflower oil [27], the fourth most consumed oil in the world [28]. Moreover, since 2013, the “silent war” of *Xylella fastidiosa* has been changing the Italian landscape, causing a decrease in the production of olive oil [29]. Therefore, alternatives to these largely consumed oils should be considered. The use of durum wheat oil in a traditional product such as *focaccia* could valorize the entire durum wheat supply chain and, at the same time, could offer producers and consumers an alternative to the currently used oils.

Therefore, the aim of this study was to evaluate the effect of durum wheat oil on the oxidation stability and physical–sensory characteristics of *focaccia*, in comparison with olive oil and sunflower oil.

## 2. Materials and Methods

### 2.1. Materials

Durum wheat oil, prepared as reported in Squeo et al. [25], was provided by Casillo Next Food Srl (Corato, Italy). Wheat flour type 0 (Casillo Spa, Corato, Italy) (carbohydrate 72 g/100 g; proteins 11 g/100 g; fat 2 g/100 g; fiber 2 g/100 g); sunflower oil (Olearia De Santis, Bitonto, Italy); olive oil (Olearia De Santis, Bitonto, Italy); yeast (*Saccharomyces cerevisiae,* Mulino Caputo, Naples, Italy); and sea salt (Atisale Spa, Margherita di Savoia, Italy) were purchased from local retailers.

### 2.2. Sample Preparation

Three different types of *focaccia* were prepared: (i) *focaccia* with sunflower oil (SO); (ii) *focaccia* with olive oil (OO); (iii) *focaccia* with durum wheat oil (DWO), according to the formulation reported in Table 1. The *focaccia* samples were prepared as described in Pasqualone et al. [12]. Flour, water, and yeast were kneaded for 6 min using a spiral kneader (Bosh MFQ40304, München, Germany). Then, salt and oil (70% of the total oil amount) were added, and kneading was continued for 6 min. The first fermentation was carried out for 1 h and 30 min in controlled conditions at 35 °C, RH = 20% (Memmert proofer, EN.CO. Srl, Spinea, Italy). The leavened dough was divided into portions, which were manually shaped into discs with a thickness of 1.5 cm and a diameter of about 30 cm. The discs were placed into metal pans, previously greased with oil (10% of the total oil amount), and left to rise again in the same conditions. The *focaccia* surface was then greased by pouring oil over it (20% of the total oil amount), followed by baking in an electric oven (Oem Ali Group Srl, Bozzolo, Italy) at 200 °C for 25 min.

### 2.3. Determination of the Resistance to Oxidation

A RapidOxy oxidation stability tester (Anton Paar, Blankenfelde-Mahlow, Germany) was used, as described in AOCS Method Cd 12c-16 [30]. Two grams of the samples (*focaccia,* finely crushed, or oil) was oxidized at a temperature of 140 °C with an oxygen pressure of 700 kPa until the pressure decreased by 10%. The samples were tested in triplicate.

### 2.4. Determination of Fatty Acid Composition

The fatty acid composition of the oils used in the *focaccia* preparation was analyzed as described by Squeo et al. [25]. A gas chromatograph (mod. 7890A, Agilent Technologies, Santa Clara, CA, USA) equipped with an FID detector (set at 220 °C) and an SP2340 capillary column of 60 m × 0.25 mm × 0.2 mm film thickness (Supelco Park, Bellefonte, PA, USA) was used to separate the fatty acid methyl esters. Comparison with the retention time of the standard mixture (C_4_–C_24_) (Sigma-Aldrich, St. Louis, MO, USA) was used for the identification of each fatty acid present in the sample. The analyses were carried out in triplicate.

### 2.5. Lipid Extraction

The lipid fraction of the *focaccia* was extracted by the Soxhlet apparatus (SER 148 extraction system, Velp Scientifica Srl, Usmate, Italy). The solvent used for the extraction was diethyl ether (Carlo Erba, Milan, Italy).

### 2.6. Determination of Tocopherols and Tocotrienols

The tocopherols and tocotrienols of oils and of the lipid fraction extracted were determined by HPLC (Agilent 1100 Series, Agilent Technologies, Santa Clara, CA, USA). Primarily, 0.02–0.03 g of sample was dissolved in 1 mL of 2-propanol. The samples were filtered by a 0.45 µm polytetrafluoroethylene (PTFE) filter and injected into an HPLC system consisting of a Waters 600E quaternary pump (Milford, MA, USA), a 7725i Rheodyne injector (20-μL sample loop), and a fluorescent detector (excitation wavelength 292 nm, emission wavelength 330 nm). The stationary phase was an Acclaim™ 120 Å C18 column, with a particle size of 3 µm and 3 mm × 150 mm in length (Thermo Fisher Scientific, Waltham, MA, USA); the mobile phase was 96:4 (*v*/*v*) methanol and water at a flow rate of 1 mL/min. The software used was Chromeleon (Thermo Fisher Scientific, Waltham, MA, USA). The single tocol was determined by the external standard method based on a previously set calibration curve. The content of tocopherols and tocotrienols was expressed as mg/kg of the total weight of oil. The analyses were carried out in triplicate.

### 2.7. Determination of Polar Compounds of the Lipid Fraction of Focaccia

The polar compounds of the oil extracted from *focaccia* samples were separated by silica gel column chromatography and quantified using high-performance size-exclusion chromatography (HPSEC) according to Difonzo et al. [31]. The content of polar compounds was expressed as g/100 g of oil extracted. The analyses were carried out in triplicate.

### 2.8. Determination of Antioxidant Activity

The extraction of antioxidant compounds was conducted as described by Troilo et al. [32], with the only changes being that the ratio of sample and extraction solvent and the number of washes were altered from 1:5 and two, respectively.

The sample extracts were submitted to a radical scavenging assays using 2,2′-azino-bis-3-ethylbenzthiazoline-6-sulphonic acid (ABTS) and 1,1-diphenyl-2-picrylhydrazyl (DPPH) radical, according to Difonzo et al. [33]. The absorbance was read using a Cary 60 spectrophotometer (Cernusco, Milan, Italy). The results were expressed as µmol Trolox equivalents (TE)/g. The determinations were carried out in triplicate.

### 2.9. Determination of Volatile Compounds

Volatile compounds of the *focaccia* were analyzed by headspace solid-phase micro-extraction (HS-SPME) coupled with gas chromatography/mass spectrometry (GC–MS), as described by Difonzo et al. [31]. The identification of the volatile compounds was carried out using an LRI and by computer matching with the reference mass spectra of the National Institute of Standards and Technology (NIST) and Wiley libraries. The quantification of the volatile compounds was performed considering the standardization of the respective peak areas with the peak area of the 1-propanol used as internal standard. The results were expressed as µg/g of sample. The analyses were carried out in triplicate.

### 2.10. Texture Profile Analysis

Texture profile analysis (TPA) was executed as described in Pasqualone et al. (2019) [12], with the only modification being the use of a cylindrical probe (36 mm diameter). The following parameters were calculated from the TPA graphic: hardness (N), chewiness (N), cohesiveness, and springiness. Six replications were carried out.

### 2.11. Color Determination

The color of *focaccia* (crumb and crust) was measured using the CM-600d colorimeter (Konica Minolta, Tokyo, Japan) supported by SpectraMagic NX software (Konica Minolta, Tokyo, Japan). The color properties were determined in the CIE (International Commission on Illumination) color space. Lightness (*L**, from black to white), red index (*a**, from green to red), and yellow index (*b**, from blue to yellow) were determined. The total color difference (∆E) was calculated as follows:$$\Delta E=\sqrt{[{\left(L^{*}-L_{0}^{*}\right)}^{2}+{\left(a^{*}-a_{0}^{*}\right)}^{2}+{\left(b^{*}-b_{0}^{*}\right)}^{2}}$$
where *L**^*^*_0_, *a**^*^*_0_, and *b**^*^*_0_ are the color coordinates of *focaccia* with olive oil (OO), while *L**, *b**, and *a** are the color coordinates of the other *focaccia* samples. The calculation considered the mean values. The following ∆E scale was considered for the evaluation of the results: 0–0.5 = no relevant difference; 0.5–1.5 = a slight difference; 1.5–3.0 = difference recognizable by an experienced observer; 3.0–6.0 = an appreciable difference; 6.0–12.0 = a large difference; and >12.0 = a very evident difference [14].

Nine replications were carried out.

### 2.12. Determination of Dimensional Parameters

The diameter (D) and thickness (T) of *focaccia* before and after baking were determined as described in Pasqualone et al. [12]. A caliper was used. The percentage variation due to baking was calculated as follows:$$\%\;\mathrm{of}\;\mathrm{variation}\;\mathrm{of}\;D\;\left(\mathrm{or}\;T\right)=\frac{\left[D\;\left(\mathrm{or}\;T\right)\;\mathrm{after}\;\mathrm{baking}\;-\;D\;\left(\mathrm{or}\;T\right)\;\mathrm{before}\;\mathrm{baking}\right]}{D\;\left(\mathrm{or}\;T\right)\;\mathrm{before}\;\mathrm{baking}\;}\times 100$$

The analyses were carried out in triplicate.

### 2.13. Determination of Sensory Properties

The quantitative descriptive analysis (QDA) of *focaccia* was conducted according to the International Standardization Organization (ISO) standard 13299 [34] by a trained panel of eight members, previously selected for their reliability, consistency, and discriminating ability. The panel was composed of four men and four women, ranging in age from 23 to 55 years, who expressed written consent according to the ethical guidelines of the laboratory of Food Science and Technology of the Department of Soil, Plant, and Food Science of the University of Bari (Italy). The panelists were regular consumers of baked products and had no food allergies or intolerances. A pre-test session was conducted, as outlined in the ISO Standard 11132 [35]. A total of 12 descriptors were selected for the consideration of the *focaccia* samples made with different oils: 3 descriptors for visual appearance (surface color intensity, inner color intensity, crumb porosity); 3 descriptors for the odor (focaccia odor, roasted odor, oxidized odor); 3 for texture perceived during the tasting (crumb elasticity, softness, crumb moisture); and 3 for taste (saltiness, sweetness, greasiness). The intensity of every attribute was expressed on a 10 cm unstructured linear scale (contractual units—c.u.). The scale anchors for focaccia odor, roasted odor, oxidized odor, saltiness, sweetness, and greasiness were: 0 c.u. = minimum intensity; 10 c.u. = maximum intensity. The scale anchors for surface and inner color intensity were: 0 c.u. = ivory; 10 c.u. = brown. The scale anchors for the crumb porosity were: 0 c.u. = dense structure with very few pores; 10 c.u. = open structure, very porous. The scale anchors for crumb moisture were: 0 c.u. = dry; 10 c.u. = wet. The scale anchors for crumb softness were: 0 c.u. = very hard; 10 c.u. = very soft. The scale anchors for crumb elasticity were: 0 c.u. rigid =; 10 c.u. = very elastic.

The samples were randomized and presented to the panelists in white dishes marked with alphanumeric codes. The testing was performed at ambient room temperature (20 ± 2 °C). In accordance with the ISO 8589 [36] standard, the sensory analysis was carried out by physically separating panelists during analysis. Three replicates were carried out.

### 2.14. Statistical Analysis

Statistical analysis was carried out by Minitab Statistical Software (Minitab Inc., State College, PA, USA). The results were all expressed as mean ± standard deviation (SD). The Anderson–Darling test was applied to evaluate the normal distribution of the data, and the Levene test was used to evaluate the homoscedasticity of variances. The significant differences (α = 0.05) were verified through the application of parametric one-way analysis of variance (ANOVA), followed by the Tukey HSD test, considering the type of oil as the independent variable.

## 3. Results and Discussion

### 3.1. Oxidation Stability, Fatty Acid Composition, and Tocols Content

Lipid oxidation is a negative event affecting many food products, particularly when their fat content is relevant and the processing or storage conditions are favourable to degradative reactions. Oxidative events cause a change in taste, texture, and appearance, as well as the production of toxic compounds and the loss of nutritional value [26].

A RapidOxy oxidation stability tester was used to evaluate the effect of varying the oil type on the oxidative stability of the *focaccia* samples. This instrument, which does not need solvents, enforces pro-oxidising conditions and measures the induction time (IT) of the lipid fraction, which is known to be positively related to the resistance to oxidation [37]. The OO *focaccia* showed the highest IT, followed by DWO and SO (Table 2).

This trend mirrored the differences observed for the oils, which, in turn, could be explained in terms of different fatty acid composition and content of antioxidant compounds.

Polyunsaturated fatty acids (PUFAs) were the most abundant class in durum wheat oil and sunflower oil (Table 3). Particularly notable was the content of linolenic acid (n-3 PUFA) observed in durum wheat oil, accounting for 5.06 ± 0.03%, while in sunflower oil the content of linolenic acid was significantly lower (0.90 ± 0.01%). This difference is relevant because studies suggest that n-3 PUFAs reduce the risk of inflammatory and cardiovascular diseases, steatohepatitis, obesity, and diabetes [38]. Although the concentration of linolenic acid in durum wheat oil was lower than in the typical sources, such as flaxseeds, it was higher than the values reported for the majority of commonly used oils, such as sunflower oil, olive oil, corn oil, and palm oil [39,40,41].

Monounsaturated fatty acids (MUFAs) were the most represented fatty acids in olive oil, which contained an amount of oleic acid accounting for 71.0 ± 0.11%. The saturated fatty acids (SFAs) were significantly more abundant in durum wheat oil and olive oil than in sunflower oil.

As PUFAs are the most susceptible to oxidation, the observed fatty acid composition easily explains the finding that olive oil showed the highest resistance to oxidation, since it had the lowest content of PUFAs. Moreover, the higher resistance to oxidation observed in durum wheat oil compared to sunflower oil, though the former had a slightly higher PUFA content, could be attributable to its higher content of SFAs, as well as to the greater presence of antioxidant compounds, primarily tocols [25].

Tocotrienols and tocopherols are recognized as natural antioxidants typical of vegetable oils and are used as additives by the food industries to cope with the low oxidative stability of PUFAs [38]. The concentrations of tocols ascertained in the oils are shown in Figure 1A, while those of the lipid fraction extracted from the *focaccia* samples are reported in Figure 1B. Durum wheat oil and DWO *focaccia* were the richest in tocotrienols, while tocopherols were more present in sunflower oil and in the corresponding *focaccia* (SO). This difference was interesting because studies have suggested that tocotrienols exert greater antioxidant activity than tocopherols and have more relevant health benefits [42,43]. Durum wheat oil contained 1094 mg/kg of tocotrienols, significantly higher than the amount determined for sunflower oil and olive oil, wherein these antioxidants were very scarce or absent. Other authors have reported similar findings in sunflower and olive oil [44] and have observed that wheat flour provides only a minimal contribution to the content of tocotrienols in baked goods [45].

### 3.2. Polar Compounds Content

The oxidative reactions that affect the lipid fraction of food determine the formation of compounds characterized by a higher polarity than unaltered triacylglycerols. In particular, oxidized triacylglycerols (ox-TAGs) are composed of triacylglycerols with an oxidized fatty acyl group, while triacylglycerol oligopolymers (TAGPs) are obtained from the latter with bonds that generate complex molecules, such as dimers and polymers. Finally, diacylglycerols (DAGs), monoacylglycerols (MAGs), and free fatty acids (FFAs) arise from the hydrolysis of triacylglycerols, as a result of lipolytic enzyme activity and moisture [18,46]. Recently, Chen et al. [18] compared the polar compounds of peanut, rapeseed, soybean, and linseed oils in different cooking conditions, observing that unsaturated fatty acids can lead to a high level of polar compounds. The SO *focaccia*, indeed, was richer in TAGPs and ox-TAGs (Figure 2) than the OO and DWO *focaccia*. These results were strongly associated with the oxidation stability; therefore, MUFA-rich oils and/or antioxidant-rich oils are able to limit the production of potentially adverse compounds [46].

The DWO *focaccia* had a significantly higher content of DAGs than SO and OO, probably due to lipolytic events affecting the raw materials used for the extraction of oil. The milling industry should consider specific containment measures for these events [25], in spite of the fact that researchers have shown the health benefits of DAGs, especially in terms of body weight [47]. DAGs also have a function in the food industry due to their emulsifying properties [26,48,49].

The presence of polar compounds in *focaccia* has previously been reported. Delcuratolo et al. [16] studied the role of different toppings on the content of polar compounds, considering only the use of extra virgin olive oil. The type of *focaccia* topping influenced the exposition to thermal stress: potato-topped *focaccia*, which was moister than *focaccia* topped with onion and rosemary, was characterized by a less intense lipid degradation. Our study, on the other hand, highlighted that the type of oil influences the concentration of polar compounds in the final products. Our trials were conducted in the absence of toppings to avoid any interferences. However, the optimal combination of toppings and oil type could allow the dramatic limitation of the content of polar compounds in *focaccia*, thus avoiding consumer exposure.

### 3.3. Antioxidant Activity

ABTS and DPPH assays are based on the color change of a sample extract in connection with the capacity of an antioxidant to reduce a colored oxidant. The antioxidants derive from the sample, while the oxidants are in the solution that is prepared for the assay [50]. For both these assays, the DWO and OO *focaccia* showed a higher antioxidant activity than SO (Table 4). These findings reflected the high content of bioactive compounds in the durum wheat and olive oils, namely tocols in durum wheat oil and phenolic compounds in olive oil (accounting for 81.5 ± 3.15 mg GAE/kg oil—data not shown). Another study [51] compared wheat germ oil (from *T. aestivum*), sunflower oil, and olive oil. The latter, however, was chosen to be high-phenolic olive oil, accounting for 320 mg GAE/kg oil, which is remarkably high considering that the refining process reduces the levels of these compounds [52]. The authors detected the highest antioxidant activity in the high-phenolic olive oil, followed by wheat germ oil and sunflower oil [51].

### 3.4. Volatile Compounds

Bread is characterized by over 540 volatile compounds [53]: alcohols, aldehydes, esters, ketones, acids, pyrazines, furans, and sulfur compounds [54], although only a small number of them really influence the flavor profile [53]. Different volatile compounds may have different origins. Microorganisms ferment the sugars and produce ethanol, which is partly lost during baking, while some of them take part in secondary fermentation events, which lead to short-chain alcohols and fatty acids, esters, and carbonyl compounds [53,54]. The oxidation of lipids causes the production of aldehydes, such as hexanal, nonanal, octanal, heptanal, and 2-heptenal. The typical baking flavor is due to the Maillard reaction involving amino acids and sugars. The caramelization of sugars and the thermal degradation of sugars and amino acids form furans, acetic acid, acetaldehyde, and other compounds [54].

In the current study, the type of oil used in the preparation of *focaccia* significantly influenced the volatile profile (Table 5). Hexanal and nonanal, markers of lipid oxidation, were significantly higher in SO, followed by DW and OO, mirroring the other chemical determinations. Additionally, the 2-methylbutanal content varied among the different oils: SO and DWO were richer in 2-methylbutanal than OO, while the content of 3-methylbutanal was higher in DWO than in SO and OO. These compounds, due to the Maillard reaction [55], positively influence the aroma of the crust, conferring a malty and roasted odor [54]. Several authors have described the effect of the fatty acid composition on the intensity of the Maillard reaction and have found that its development is favored by a higher unsaturation level [56].

The Maillard reaction also generated benzaldehyde and furans; the content of the former was significantly lower in OO, while the content of the latter was significantly higher in DWO. This result could be attributed to the simultaneous presence, in DWO, of high concentrations of PUFA and diglycerides, which positively influence the presence of furans, as observed by Emektar et al. [57]. Pyrazines, with their olfactory properties, confer a pleasant roasted odor on baked goods [54,58] and are therefore used as additives to improve the organoleptic properties of bread and other bakery products [58]. DWO was significantly richer in pyrazines than SO and OO. These findings could be connected to the differing acidic compositions of the oils used, in particular to the PUFA content, which was the highest in durum wheat oil, followed by sunflower oil, then olive oil. In support of this, Negroni et al. [59], studying the formation of pyrazines in glucose–lysine or xylose–lysine model systems added to olive oil, canola oil, and sunflower oil, suggested that higher unsaturation levels could lead to a higher presence of pyrazines.

The fermentation of *focaccia*, caused by compressed yeast (*Saccharomyces cerevisiae*), produces ethyl alcohol. Despite its partial evaporation during baking, ethyl alcohol contributes to the aroma of baked goods [60], but its concentration was not influenced by the type of oil.

### 3.5. Physical Determinations

Ingredients and processing, especially baking, are principally responsible for the color of baked products: the golden-brown color of the crust is considered an important quality parameter [61]. Table 6 reports the colorimetric indices of the *focaccia* prepared with different oils, shown in Figure 3. DWO crumb and crust were significantly less luminous (lower *L**) and showed higher *a** (more intense red tone) than SO and OO, while no significant differences were observed for *b** (yellow index). These results agreed with the data for volatile compounds. In fact, higher levels of pyrazines and furans, both arising from thermal reactions which cause browning, were observed in DWO than in the other two *focaccia* types. These observations were reinforced by the calculation of the total color differences (∆E) of the crust and crumb, with OO taken as reference. The color differences of the crumb were lower than those observed in the crust, with the latter being more exposed to heat and more impacted by non-enzymatic browning. In particular, the difference in color between OO and DWO crumbs was considered recognizable only by an experienced observer (1.5 < ∆E < 3.0). Instead, the difference in crust color was considered clearly recognizable (3.0 < ∆E < 6.0). Other authors, studying bread, have reported the effect of the type of oil on the color of the crumb and crust [62,63].

Texture profile analysis (TPA) consisted of compressing a food sample twice in a reciprocating motion that mimicked the action of the jaw [64]. Four parameters were measured: hardness, springiness, chewiness, and cohesiveness. The springiness and cohesiveness were very similar in all *focaccia* types, while the hardness and chewiness showed significant differences among samples (Table 6). In particular, the use of durum wheat oil and olive oil were related to a lower hardness and chewiness than sunflower oil. This result could be related to the content of DAGs, which was higher in these two oils than in the sunflower oil. DAGs, indeed, together with monoglycerides, are extensively used in breadmaking as emulsifiers to improve crumb softness. In addition, their presence can delay the firming process, due to the ability to form complexes with amylose and amylopectin [49,65].

During baking, an increase in volume occurs due to the thermal expansion of gases [12]. As a consequence, the thickness of the *focaccia* increased with baking, to a similar extent in all samples (Table 6). Meanwhile, the diameter decreased, but without an influence exerted by the type of oil. The effect of oil on the variation in the dimensional parameters during baking, therefore, was secondary, while other studies have reported a significant effect caused by the type of flour, due to differing fiber and gluten contents [12].

### 3.6. The Sensory Profile of Focaccia

The type of oil also significantly influenced the sensory properties of *focaccia* (Table 7).

The perception of crumb and crust color varied with the type of oil, with DWO being darker than the others. The sensory evaluation of color agreed with the instrumental determination. Moreover, DWO was perceived as softer and more porous than the other *focaccia* types, while the elasticity of the crumbs was similar in all samples.

The type of oil did not affect the perception of sweetness and crumb moisture, while the panelists perceived DWO to be less salty and oily, which was interesting, considering the preference of consumers for *focaccia* that has not been excessively greased [12]. A hint of oxidized odor was detected only in SO, while none was observed in DWO and OO. A roasted odor, as well as the typical odor of *focaccia*, were both perceived significantly more intensely in DWO, due to its higher content of pyrazines.

## 4. Conclusions

Considering the significant nutritional, sensory, and health importance of the lipid fraction of *focaccia,* this study suggested that the choice of oil to be used in its preparation is not trivial. Although olive oil, rich in MUFAs, was proven to be the most resistant to oxidation, durum wheat oil, rich in PUFAs and tocols, was more stable than sunflower oil thanks to the greater presence of antioxidants. Moreover, the use of durum wheat oil demonstrated a positive effect on the physical and sensory characteristics of the end product. Therefore, the reformulation of bakery products with this oil will increase the value of the by-products generated by the durum wheat milling industries, while respecting the principles of the circular economy. This oil could offer a healthier alternative to consumers while combining the tradition of *focaccia* making with a viable strategy for product innovation and, at the same time, increasing the sustainability of the durum wheat chain.

Durum wheat oil could also respond to the need to find new alternatives to sunflower oil, the supply of which is facing considerable difficulties due to the war in Ukraine. It should be noted, however, that durum wheat oil is a high-quality niche product with a relatively high price (5.00 €/kg, compared to 2.50–3.00 €/kg for olive oil and 1.50–2.00 €/kg for sunflower oil). Its price is justified by the high nutritional value related to the remarkably high concentration of tocols, especially tocotrienols, and favorable levels of n-3 PUFAs. Currently, there is a single producer of durum wheat oil, with a productive capacity of 4000 tons/year. Therefore, there is presently not enough durum wheat oil to make up for potential losses in olive and sunflower oil, but there is good development potential because other companies will probably start producing it in the future.

Future investigations, however, are needed to deepen our knowledge of the effect of this oil on products’ shelf lives and to widen its application in the food sector and beyond. In particular, the performance of durum wheat oil during *focaccia* storage should be investigated by conducting shelf-life studies in comparison with other refined oils. Furthermore, durum wheat oil could also be considered for interesting applications in pharmaceuticals, nutraceuticals, and the cosmetic sector, which could represent the main routes, alongside the food industry, for the valorization of cereal by-products.

## Figures and Tables

**Figure 1 foods-11-02679-f001:**
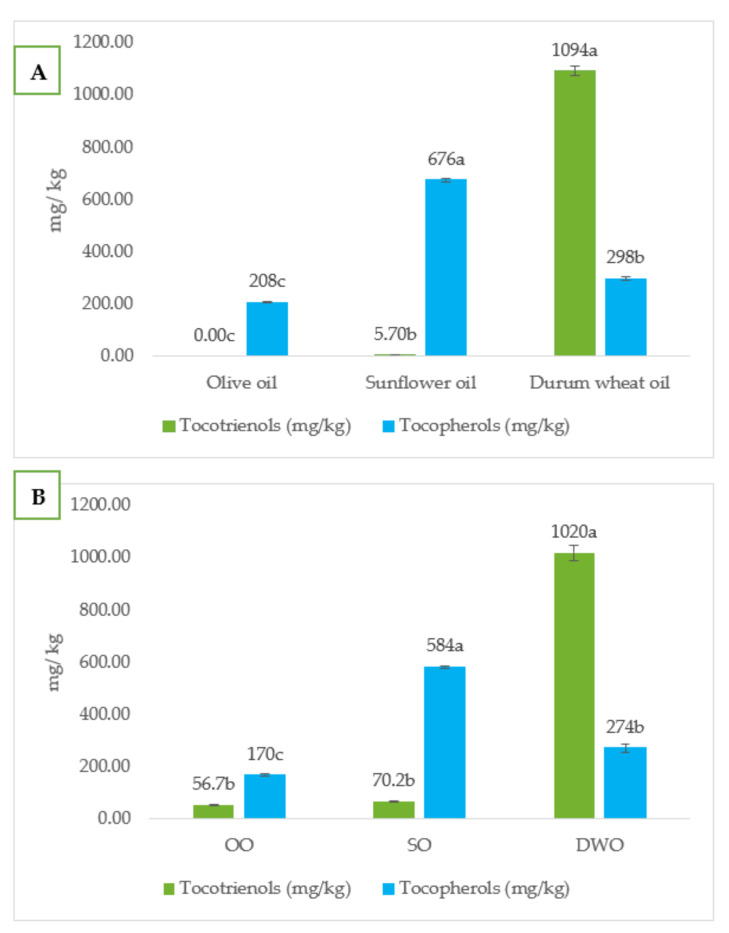
Tocopherols and tocotrienols content (mg/kg) of oil (**A**) and *focaccia* (**B**). SO = *focaccia* with sunflower oil; OO = *focaccia* with olive oil; DWO = *focaccia* with durum wheat oil. Data are presented as means ± SD of three replicates. Different letters in the same row indicate significant differences at *p* < 0.05.

**Figure 2 foods-11-02679-f002:**
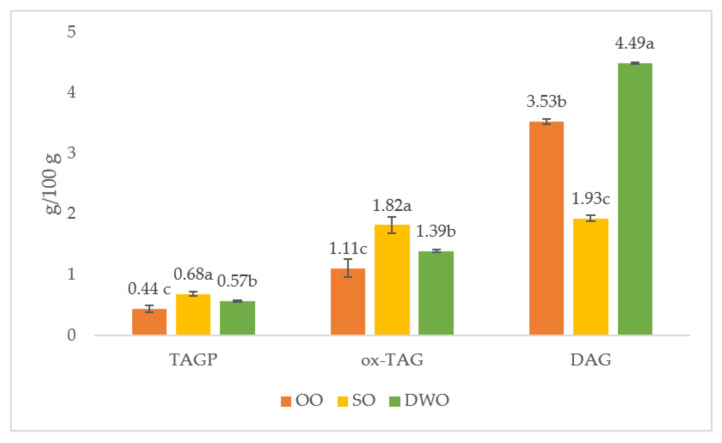
Polar compounds content (g/100 g of extracted fat) of *focaccia.* SO = *focaccia* with sunflower oil; OO = *focaccia* with olive oil; DWO = *focaccia* with durum wheat oil; TAGP = triacylglycerol oligopolymers; ox-TAG = oxidized triacylglycerols; DAG = diacylglycerols. Data are presented as means ± SD of three replicates. Different letters in the same row indicate significant differences at *p* < 0.05.

**Figure 3 foods-11-02679-f003:**
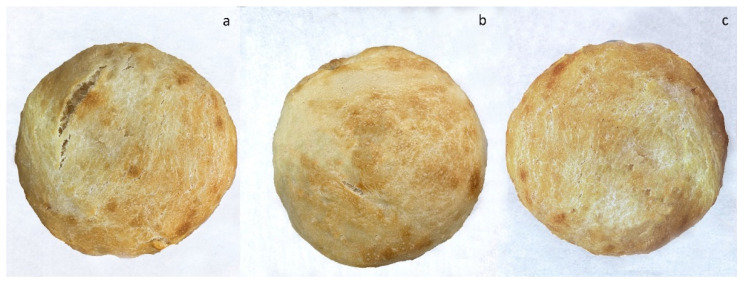
(**a**) *Focaccia* with olive oil; (**b**) *focaccia* with durum wheat oil; (**c**) *focaccia* with sunflower oil.

**Table 1 foods-11-02679-t001:** Formulation of the experimental *focaccia* samples. SO = *focaccia* with sunflower oil; OO = *focaccia* with olive oil; DWO = *focaccia* with durum wheat oil.

Ingredients	SO (g)	OO (g)	DWO (g)
Wheat flour type 0	600	600	600
Water	420	420	420
Sunflower oil	85	-	-
Olive oil	-	85	-
Durum wheat oil	-	-	85
Salt	15	15	15
Yeast	5	5	5

**Table 2 foods-11-02679-t002:** Resistance to forced oxidation of the oils and *focaccia*. SO = *focaccia* with sunflower oil; OO = *focaccia* with olive oil; DWO = *focaccia* with durum wheat oil.

Sample	Induction Time (min)
*Focaccia*	
OO	134 ± 2.06 ^a^
SO	66.3 ± 4.81 ^c^
DWO	86.2 ± 2.53 ^b^
*Oils*	
Olive oil	59.5 ± 0.07 ^a^
Sunflower oil	30.7 ± 0.64 ^c^
Durum wheat oil	39.8 ± 0.26 ^b^

Data are presented as means ± SD of three replicates. Different letters for the same sample type indicate significant differences at *p* < 0.05.

**Table 3 foods-11-02679-t003:** Percentage of the main fatty acids in the olive oil, sunflower oil, and durum wheat oil used in the preparation of *focaccia* samples.

Fatty Acids (%)	Olive Oil	Durum Wheat Oil	Sunflower Oil
C_14:0_	0.02 ± 0.00 ^c^	0.07 ± 0.00 ^b^	0.08 ± 0.00 ^a^
C_16:0_	12.9 ± 0.06 ^b^	14.84 ± 0.04 ^a^	6.48 ± 0.01 ^c^
C_18:0_	3.24 ± 0.04 ^b^	1.38 ± 0.01 ^c^	3.82 ± 0.01 ^a^
C_18:1_	71.0 ± 0.11 ^a^	22.2 ± 0.05 ^c^	30.3 ± 0.00 ^b^
C_18:2_	9.42 ± 0.08 ^c^	55.1 ± 0.05 ^b^	57.5 ± 0.01 ^a^
C_18:3_	0.90 ± 0.01 ^b^	5.06 ± 0.03 ^a^	0.23 ± 0.01 ^c^
∑SFA	16.9 ± 0.03 ^b^	17.0 ± 0.00 ^a^	10.8 ± 0.01 ^c^
∑MUFA	72.5 ± 0.11 ^a^	22.6 ± 0.06 ^c^	30.5 ± 0.00 ^b^
∑PUFA	10.6 ± 0.08 ^c^	60.5 ± 0.06 ^a^	58.7 ± 0.01 ^b^

∑SFA, sum of saturated fatty acids; ∑MUFA, sum of monounsaturated fatty acids; ∑PUFA, sum of polyunsaturated fatty acids. Data are presented as means ± SD of three replicates. Different letters in the same row indicate significant differences at *p* < 0.05.

**Table 4 foods-11-02679-t004:** Total antioxidant capacity of experimental *focaccia*. OO = *focaccia* prepared with olive oil; DWO = *focaccia* prepared with durum wheat oil; SO = *focaccia* prepared with sunflower oil.

Focaccia Type	ABTS (µmol TE/g)	DPPH (µmol TE/g)
OO	0.64 ± 0.03 ^ab^	0.46 ± 0.01 ^ab^
DWO	0.70 ± 0.04 ^a^	0.55 ± 0.02 ^a^
SO	0.56 ± 0.04 ^b^	0.38 ± 0.03 ^b^

Expressed as µmol/g trolox equivalent. Data are presented as means ± SD of three replicates. Different letters in the same column indicate significant differences at *p* < 0.05.

**Table 5 foods-11-02679-t005:** Volatile compounds of experimental *focaccia*. OO = *focaccia* prepared with olive oil; DWO = *focaccia* prepared with durum wheat oil; SO = *focaccia* prepared with sunflower oil.

Volatile Compounds (µg/g)	*Focaccia* Type		
	OO	DWO	SO
*Aldehydes*			
Hexanal	15.7 ± 0.01 ^c^	22.2 ± 0.0 ^b^	25.8 ± 1.07 ^a^
Heptanal	1.00 ± 0.08 ^b^	1.85 ± 0.02 ^a^	2.01 ± 0.24 ^a^
Nonanal	4.91 ± 0.33 ^b^	4.83 ± 0.15 ^b^	7.20 ± 0.01 ^a^
2-Methylbutanal	12.0 ± 0.45 ^c^	17.7 ± 0.64 ^b^	20.5 ± 0.31 ^a^
3-Methylbutanal	16.3 ± 0.63 ^c^	25.5 ± 0.66 ^a^	22.7 ± 0.08 ^b^
Octanal	1.35 ± 0.02 ^ab^	0.87 ± 0.16 ^b^	1.71 ± 0.12 ^a^
2-Heptenal	5.00 ± 0.01 ^b^	4.63 ± 0.11 ^c^	9.75 ± 0.15 ^a^
2,4-Heptadienal	0.78 ± 0.06 ^c^	1.56 ± 0.08 ^b^	3.28 ± 0.11 ^a^
Benzacetaldheyde	2.57 ± 0.04 ^b^	4.58 ± 0.09 ^a^	1.83 ± 0.08 ^c^
Benzaldehyde	6.18 ± 0.55 ^b^	7.55 ± 0.36 ^a^	7.28 ± 0.18 ^a^
*Alcohols*			
Ethyl alcohol	2.29 ± 0.32 ^a^	2.10 ± 0.46 ^a^	2.67 ± 0.22 ^a^
2-Phenylethanol	8.54 ± 0.16 ^a^	4.73 ± 0.12 ^c^	7.57 ± 0.23 ^b^
1-Hexanol	6.14 ± 0.00 ^b^	2.82 ± 0.13 ^c^	10.8 ± 0.08 ^a^
*Carboxylic acid*			
Acetic acid	2.95 ± 0.00 ^a^	1.41 ± 0.09 ^c^	1.73 ± 0.05 ^b^
*Ketones*			
Methyl ethyl ketone	1.72 ± 0.28 ^a^	1.73 ± 0.17 ^a^	1.48 ± 0.04 ^a^
*Furan compounds*			
2-Furanmethanol	1.28 ± 0.25 ^c^	9.67 ± 0.90 ^a^	6.59 ± 0.27 ^b^
Furan-2-pentyl	2.25 ± 0.26 ^c^	4.80 ± 0.35 ^a^	3.83 ± 0.09 ^b^
2-Furancarboxaldehyde, 5-methyl-	0.50 ± 0.10 ^c^	1.44 ± 0.07 ^a^	0.63 ± 0.04 ^b^
2-Furancarboxaldehyde (furfural)	1.46 ± 0.01 ^c^	5.32 ± 0.06 ^a^	5.15 ± 0.06 ^b^
*Pyrazines*			
Methyl-pyrazine	2.72 ± 0.11 ^c^	10.4 ± 0.82 ^a^	8.20 ± 0.85 ^b^
Ethyl-pyrazine	1.42 ± 0.12 ^c^	2.66 ± 0.07 ^a^	2.42 ± 0.02 ^b^

Data are presented as means ± SD of three replicates. Different letters in the same row indicate significant differences at *p* < 0.05.

**Table 6 foods-11-02679-t006:** Physical determinations (color, texture, and dimensional variations during baking) of the experimental *focaccia* samples. OO = *focaccia* prepared with olive oil; DWO = *focaccia* prepared with durum wheat oil; SO = *focaccia* prepared with sunflower oil.

	*Focaccia* Type		
	OO	DWO	SO
**Color**			
*Crumb*			
*a**	0.40 ± 0.08 ^b^	0.87 ± 0.11 ^a^	0.67 ± 0.08 ^a^
*b**	18.2 ± 0.31 ^a^	18.99 ± 0.46 ^a^	21.0 ± 0.38 ^a^
*L**	72.8 ± 1.37 ^ab^	71.15 ± 1.58 ^b^	74.5 ± 0.13 ^a^
∆E	-	1.88	3.28
*Crust*			
*a**	7.15 ± 0.34 ^b^	10.1 ± 1.77 ^a^	9.31 ± 0.66 ^ab^
*b**	32.0 ± 1.14 ^a^	33.4 ± 2.02 ^a^	33.2 ± 2.39 ^a^
*L**	67.5 ± 0.82 ^a^	62.7 ± 1.89 ^b^	64.7 ± 1.99 ^ab^
∆E	-	5.82	3.71
**Texture**			
Hardness (N)	7.69 ± 1.06 ^b^	8.67 ± 1.13 ^b^	12.1 ± 1.13 ^a^
Springiness	0.94 ± 0.01 ^a^	0.94 ± 0.02 ^a^	0.95 ± 0.01 ^a^
Chewiness (N)	5.73 ± 0.84 ^b^	6.39 ± 0.44 ^b^	9.80 ± 0.48 ^a^
Cohesiveness	0.79 ± 0.01 ^a^	0.82 ± 0.07 ^a^	0.82 ± 0.01 ^a^
**Dimensional variations during baking**			
Diameter variation (%)	−0.73 ± 0.01 ^a^	−0.73 ± 0.01 ^a^	−0.75 ± 0.01 ^a^
Thickness variation (%)	117 ± 9.91 ^a^	110 ± 11.7 ^a^	118 ± 8.08 ^a^

Data are presented as means ± SD of three replicates. Different letters in the same row indicate significant differences at *p* < 0.05.

**Table 7 foods-11-02679-t007:** Sensory profile of experimental *focaccia*. OO = *focaccia* prepared with olive oil; DWO = *focaccia* prepared with durum wheat oil; SO = *focaccia* prepared with sunflower oil.

	*Focaccia* Type		
Sensory Descriptor	OO	DWO	SO
Surface color	3.80 ± 0.35 ^b^	5.55 ± 0.28 ^a^	4.22 ± 0.30 ^b^
Inner color	0.58 ± 0.12 ^b^	0.85 ± 0.05 ^a^	0.75 ± 0.00 ^a^
Crumb porosity	4.27 ± 0.35 ^b^	5.53 ± 0.12 ^a^	3.57 ± 0.25 ^b^
*Focaccia* odor	6.50 ± 0.05 ^b^	7.67 ± 0.25 ^a^	6.75 ± 0.05 ^b^
Oxidized odor	0.00 ± 0.00 ^b^	0.00 ± 0.00 ^b^	0.63 ± 0.10 ^a^
Roasted odor	1.15 ± 0.15 ^b^	1.67 ± 0.22 ^a^	1.13 ± 0.15 ^b^
Crumb elasticity	5.38 ± 0.06 ^a^	5.17 ± 0.38 ^a^	5.60 ± 0.22 ^a^
Softness	6.18 ± 0.29 ^b^	7.08 ± 0.08 ^a^	5.77 ± 0.08 ^b^
Crumb moisture	5.55 ± 0.05 ^a^	5.55 ± 0.79 ^a^	5.48 ± 0.08 ^a^
Greasiness	6.07 ± 0.19 ^a^	5.12 ± 0.43 ^b^	5.93 ± 0.19 ^a^
Sweetness	1.15 ± 0.13 ^a^	1.22 ± 0.28 ^a^	1.42 ± 0.08 ^a^
Saltiness	5.03 ± 0.20 ^a^	4.37 ± 0.25 ^b^	4.90 ± 0.09 ^a^

Data are presented as means ± SD of three replicates. Different letters in the same row indicate significant differences at *p* < 0.05.

## Data Availability

Data is contained within the article.

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
