# Peer review of "The Use of Durum Wheat Oil in the Preparation of *Focaccia*: Effects on the Oxidative Stability and Physical and Sensorial Properties"

_foods, 2022, doi:10.3390/foods11172679_

Round 1

Reviewer 1 Report

Interesting paper with timely relevance. Is there enough production of durum oil to make up for potential losses in olive and sunflower oil? What is the price comparison?

Other comments:

The sensory work is described very well!!

Nearly 100 references seems excessive,  38 in the introduction. Perhaps aim for 50.

Line 112: RapidOxy (Anton Paar, Blankenfelde-Mahlow, Germany) was used to test the oxida add Oxidation Stability Tester  e.g. RapidOxy oxidation stability tester..

Reference the approved standard methood: AOCS Method Cd 12c-16 “Accelerated Oxidation Test for the Determination of the Oxidation Stability of Foods, Oils and Fats Using the Oxitest Oxidation Test Reactor”

 For GC what standards were used?

What temperature was used for lipid extraction

Line 152 change “par” to “from”

Line 241       Oxytest was used to quickly evaluate the effect of the vegetable oil on the oxidative stability of focaccia samples. This test, which does not need solvents, forces the pro-oxidising 242 conditions and measures the induction time (IT), which is known to be positively related to the oxidation resistance [47-49].

This should be clarified – OXITEST or the RapidOxy  instrument.  

Figure 1 please split into separate sub figures oils (1a) and focaccia (1b)

Author Response

Response to Revewer 1.

Interesting paper with timely relevance.

Answer: We thank the reviewer for his/her positive evaluation of our work. We implemented all your suggestions. Modifications are made in red (while the blue ones have a linguistic nature).

Is there enough production of durum oil to make up for potential losses in olive and sunflower oil? What is the price comparison?

Answer: Thanks for these interesting questions. Very helpful in giving a practical perspective to the study. Durum wheat oil is a high quality niche product with a relatively high price (5.00 €/kg, compared to 2.50-3.00 €/kg for olive oil and 1.50-2.00 €/kg for sunflower oil). Its price is justified by the high nutritional value related to the remarkably high concentration of tocols, especially tocotrienols, and interesting levels of n-3 PUFAs. Currently, there is a single producer of durum wheat oil, with a productive capacity of 4000 tons/year. Therefore, there is still not enough durum wheat oil to make up for potential losses in olive and sunflower oil, but there is good development potential because other companies will probably start producing it in the future.

We added these considerations in the Conclusive section (see lines 457-464).

Other comments:

The sensory work is described very well!!

Answer: We thank very much the reviewer for his/her positive evaluation.

Nearly 100 references seems excessive,  38 in the introduction. Perhaps aim for 50.

Answer: The number of references has been reduced.

Line 112: RapidOxy (Anton Paar, Blankenfelde-Mahlow, Germany) was used to test the oxida add Oxidation Stability Tester  e.g. RapidOxy oxidation stability tester.

Answer: Thank you very much for noting. The sentence has been amended modified (line 106).

Reference the approved standard methood: AOCS Method Cd 12c-16 “Accelerated Oxidation Test for the Determination of the Oxidation Stability of Foods, Oils and Fats Using the Oxitest Oxidation Test Reactor”

Answer: Thank you very much for noting. The method AOCS has been added in the references.

 For GC what standards were used?

Answer: Thank you very much for noting. The sentence has been amended (line 116).

What temperature was used for lipid extraction

Answer: The Soxhlet apparatus applied a temperature of 90°C for the lipid extraction.

Line 152 change “par” to “from”

 Answer: Thank you very much for noting. The sentence has been amended (line 145).

Line 241       Oxytest was used to quickly evaluate the effect of the vegetable oil on the oxidative stability of focaccia samples. This test, which does not need solvents, forces the pro-oxidising 242 conditions and measures the induction time (IT), which is known to be positively related to the oxidation resistance [47-49].

This should be clarified – OXITEST or the RapidOxy  instrument.  

Answer: Thank you very much for noting. The sentence has been amended (line 230)

Figure 1 please split into separate sub figures oils (1a) and focaccia (1b)

Answer: Thank you very much for noting. The figures have been separated into sub figures (1a and 1b).

Reviewer 2 Report

My comments are as follow:

·         Page 1 line 22. Since some words are repeated in the manuscript title and in the keyword section, I suggest to replace the words: “focaccia” or “durum wheat” by “antioxidant activity”, “volatile compounds”, “texture profile analysis” or other keywords, to have more visibility of your manuscript paper.

·         In the determination of sensory properties section, what king of statistical analysis did you applied ? parametric or non-parametric analysis? The data obtained with the eight panellist (four men and four women) results, was analysed in relation to the homoscedasticity of variances, independence of errors, and normally of the residuals?

·         In the Statistical Analysis section. For the research data, did you apply parametric or non-parametric analyses? Did you analyse the homoscedasticity of variances, independence of errors, and if residuals are normally distributed? I suggest to include more information about those analyses and the abovementioned section.

·         In the Statistical Analysis section, line 231. I suggest to include the significance level is 0.05 (α=0.05) applied on your data.

·         In all the Tables and numbers of your manuscript, I suggest to work with two decimals for numbers with one number before the “decimal point” (for example, in Table 3, 3.24 it is OK), with one decimal for numbers with two numbers before the “decimal point” (for example, in Table 2, round to 66.3 instead of 66.33), and with no decimals for numbers with three or more numbers before the “decimal point” (for example, in Table 2, round to 134 instead of 134.07). I suggest to do the same with all the numbers and table, as appropriate in your manuscript.

·         Regarding the presentation of Gumminess and Chewiness data, some literature related to texture profile methodology said that chewiness is for solid foods and gumminess is for semisolid foods so, is it correct to show both?

·         In the conclusion section, I found information related to the Use of Durum Wheat Oil in the preparation of Focaccia: Tradition, Innovation, and Sustainability. Nevertheless, what about the future work in your research? I suggest to state with more accurate future research.

Author Response

Response to Reviewer 2

We thank the reviewer for careful reading. We implemented all the Reviewer’s suggestions. Modifications are made in red (while the blue ones have a linguistic nature).

Page 1 line 22. Since some words are repeated in the manuscript title and in the keyword section, I suggest to replace the words: “focaccia” or “durum wheat” by “antioxidant activity”, “volatile compounds”, “texture profile analysis” or other keywords, to have more visibility of your manuscript paper.

Answer: Thank you very much for your suggestion. The words “focaccia” has been replaced with “texture profile analysis” (line 23)

In the determination of sensory properties section, what king of statistical analysis did you applied ? parametric or non-parametric analysis? The data obtained with the eight panellist (four men and four women) results, was analysed in relation to the homoscedasticity of variances, independence of errors, and normally of the residuals?

Answer: Thanks for the helpful suggestions. The results of sensory properties were analysed as described in the revised version of Statistical Analysis section, modified after your suggestions.

In the Statistical Analysis section. For the research data, did you apply parametric or non-parametric analyses? Did you analyse the homoscedasticity of variances, independence of errors, and if residuals are normally distributed? I suggest to include more information about those analyses and the abovementioned section.

Answer: Thanks for helpful suggestions. The information have been included in the revised version of the Statistical Analysis section.

In the Statistical Analysis section, line 231. I suggest to include the significance level is 0.05 (α=0.05) applied on your data.

Answer: Thanks for the helpful suggestion. The information have been included in the revised version of the Statistical Analysis section.

In all the Tables and numbers of your manuscript, I suggest to work with two decimals for numbers with one number before the “decimal point” (for example, in Table 3, 3.24 it is OK), with one decimal for numbers with two numbers before the “decimal point” (for example, in Table 2, round to 66.3 instead of 66.33), and with no decimals for numbers with three or more numbers before the “decimal point” (for example, in Table 2, round to 134 instead of 134.07). I suggest to do the same with all the numbers and table, as appropriate in your manuscript.

Answer: Thank you very much for noting. Decimals have been modified.

Regarding the presentation of Gumminess and Chewiness data, some literature related to texture profile methodology said that chewiness is for solid foods and gumminess is for semisolid foods so, is it correct to show both?

Answer: Thank you very much for noting. We have modified, considering only the chewiness and not gumminess.

In the conclusion section, I found information related to the Use of Durum Wheat Oil in the preparation of Focaccia: Tradition, Innovation, and Sustainability. Nevertheless, what about the future work in your research? I suggest to state with more accurate future research.

Answer: Thanks for the helpful suggestions. The future studies and applications have been added in the conclusion section (see lines 465-471).

Author Response

Response to Reviewer 3

In manuscript entitled ‘The Use of Durum Wheat Oil in the preparation of Focaccia: Tradition, Innovation, and Sustainability’ authors reported use of DWO as an alternative for OO and SO with better sensory and nutritional profile. Presented work is well organized, with comprehensively explained methodology and results. However, following points need consideration.

We thank the reviewer for careful reading, and for positive evaluating our work. We implemented all the Reviewer’s suggestions. Modifications are made in red, while the linguistic ones are blue.

Comment 1:

I will suggest change in title as the current title is very general and broad. Make it more specific to the study objectives.

Answer: We modified the title to make it more specific, as follows: “The Use of Durum Wheat Oil in the Preparation of Focaccia: Effects on the Oxidative Stability, Physical and Sensorial Properties”.

Comment 2:

Manuscript need language editing throughout the document, as at many places it’s difficult to comprehend the meaning, few examples are here L-20, L-33, L-40, L-45, L-49, L-61-65, L-267-27, L-3391, L-400, L-403 L-412, L- 433, L- 449, L-454.

Answer: Sorry, language editing has been made in every point you raised and throughout the entire manuscript. Please see the blue modifications.

Comment 3:

For sensory analysis, can you justify the number of penal members? Is the panel of 8 judges sufficient for this analysis? How many replicates were performed?  Kindly justify.

Answer: Thank you very much for noting. The number of 8 judges is sufficient for this analysis and in line with the number generally used, because they are trained panelists. In addition, as described by the ISO rules, the number of judges is also connected with the type of test conducted. For the QDA analysis the number is between 6 and 20 (doi:10.1016/B978-0-12-382086-0.00006-6); the specific number depends on the reliability, consistency, and discriminating ability of candidates and their training. We added a sentence (line 193) to specify that panelists (which were trained, as already specified) were previously selected for their reliability, consistency, and discriminating ability. The number of replicates is detailed at line 215.

Comment 4:

Would be interested to know economic impact of DWO usage compared to OO and SO.

Answer: Durum wheat oil is a high quality niche product with a relatively high price (5.00 €/kg, compared to 2.50-3.00 €/kg for olive oil and 1.50-2.00 €/kg for sunflower oil). Its price is justified by the high nutritional value related to the remarkably high concentration of tocols, especially tocotrienols, and interesting levels of n-3 PUFAs. We added these considerations in the Conclusion section (see lines 457-464).

Comment 5:

What about shelf life? Did you performed this comparison among Focaccia prepared from three different oils?

Answer: The shelf life has not been evaluated, but thanks for your suggestion. The comparison could be very interesting in future work. This is described in the modified conclusion section (Line 465-468).

Comment 6:

In antioxidant assays, antioxidant contents were higher in DWO compared to two other oils, but resistance to forced oxidation was less compared to OO, can you kindly explain this?

Answer: As presented in section 3.1, the resistance of olive oil is not only due to antioxidant compounds but also to the acidic composition. The MUFAs confer higher resistance to olive oil than durum wheat oil, rich in PUFAs. Furthermore, the olive oil used in the trials contained also other antioxidants, namely phenolic compounds (as it was stated at line 19 of the Abstract and lines 331-332 of Results and discussion).

Comment 7:

Analysis of fatty acid compositions (L-124) and Volatile compounds (L-166) was done in duplicate, Why not in triplicate? Moreover, in legends (tables and figures) information regarding number of replicates and whether presented values are ± SD or SE is missing.

Answer: Thank you very much for noting. In the volatile compounds and in the fatty acid composition there were typo-it because the analyses were carried out in triplicate. The tables and figures have been corrected.

Comment 8:

Though Methodology section is well explained, yet need to remove plagiarism. Kindly work on this end.

Answer: Thank you very much for noting. The Methodology section has been reworded to remove plagiarism.
